# *Bifidobacterium longum* and *Chlorella sorokiniana* Improve the IFN Type I-Mediated Antiviral Response in Rotavirus-Infected Cells

**DOI:** 10.3390/microorganisms11051237

**Published:** 2023-05-08

**Authors:** Ricardo Romero-Arguelles, Patricia Tamez-Guerra, Guadalupe González-Ochoa, César I. Romo-Sáenz, Ricardo Gomez-Flores, Lilian Flores-Mendoza, Elizama Aros-Uzarraga

**Affiliations:** 1Laboratorio de Inmunología y Virología, Facultad de Ciencias Biológicas, San Nicolás de los Garza, Universidad Autónoma de Nuevo León, San Nicolás de los Garza 66455, Mexico; ricardoromeroarguelles@gmail.com (R.R.-A.); cirscirscirs1989@gmail.com (C.I.R.-S.); rgomez60@hotmail.com (R.G.-F.); 2Departamento de Ciencias Químico Biológicas y Agropecuarias, Universidad de Sonora, Navojoa 85880, Mexico; lilian.flores@unison.mx (L.F.-M.); a215216586@unison.mx (E.A.-U.)

**Keywords:** probiotics, immunity, *Bifidobacterium*, *Chlorella*, rotavirus, gastroenteritis

## Abstract

Probiotics are effective to treat or prevent gastrointestinal infections, and microalgae have demonstrated important health-promoting effects and in some cases function as prebiotics. In this regard, the anti-rotavirus effect of *Bifidobacterium longum* and *Chlorella sorokiniana* by reducing viral infectivity is well known. However, their effect on immune response against rotavirus has not yet been investigated. Therefore, the aim of this study was to determine the role of *Bifidobacterium longum* and/or *Chlorella sorokiniana* in influencing an IFN type I-mediated antiviral response in rotavirus-infected cells. In pre-infection experiments, HT-29 cells were treated with *B. longum* and *C. sorokiniana* alone or in combination, followed by rotavirus infection, whereas in post-infection assays, HT-29 cells were treated after infection. The cells’ mRNA was then purified to determine the relative expression level of IFN-α, IFN-β, and precursors of interferons such as RIG-I, IRF-3, and IRF-5 by qPCR. We showed that combination of *B. longum* and *C. sorokiniana* significantly increased IFN-α levels in pre-infection and IFN-β in post-infection assays, as compared with individual effects. Results indicate that *B. longum*, *C. sorokiniana*, or their combination improve cellular antiviral immune response.

## 1. Introduction

Probiotics are commensal microorganisms that colonize human gut, modulate microbiota, and boost host immunity, improving health [1,2]. Some probiotics, such as *Lactobacillus* and *Bifidobacterium* species, are commonly used because they have effectively treated or prevented gastrointestinal diseases such as infectious gastroenteritis [3]. Their beneficial effect is attributed to the production of antimicrobial metabolites, including short-chain fatty acids, bacteriocins, reuterin, linoleic acid, and secondary bile acids, as well as their potential to enhance intestinal or systemic immunity [4,5], particularly the cellular immune response [6,7,8]. Furthermore, *Bifidobacterium* species have been shown to inhibit virus replication, particularly rhinovirus, herpes simplex virus, coxsackievirus, human papillomavirus, noroviruses, and hepatitis B virus [6,8,9].

Despite probiotics’ beneficial properties, their effectiveness is conditioned to viability and abundance at the time of consumption (at least 10^6^ CFU/g are potentially effective) [10]. To improve probiotics’ viability, some researchers made combinations with prebiotics or microalgae such as *Chlorella* spp. [1,9]. Some studies with *Chlorella sorokiniana* and *C. vulgaris* have shown prebiotic activity by enhancing *Lactobacillus rhamnosus*, *L. acidophilus*, and *Bifidobacterium lactis* growth and viability [11,12,13]. Furthermore, *C. sorokiniana* has been associated with improved shelf-life of *B. longum* and *L. plantarum* [9,14]. Although *Chlorella* is not yet formally defined as prebiotic, it is considered a functional food [14,15]. In vitro assays with *Chlorella* showed an increased propionate-producing microorganisms’ population in the intestines, which was associated with postbiotic gut health [16]. Moreover, *Chlorella* has shown gastrointestinal effects in the management of mild digestive disorders and the fecal microbiota in pigs. It has also demonstrated antimicrobial activity against intestinal pathogens [11,14,15,17,18].

On the other hand, rotavirus infection is one of the main causes of gastroenteritis. Despite vaccination, it is the leading cause of diarrhea-associated morbidity and mortality in children up to five years old worldwide, particularly in developing countries [19,20]. Rotavirus gastroenteritis is characterized by watery diarrhea, vomiting, fever, and dehydration. The clinical course varies from mild to severe, sometimes leading to death. Symptoms are more severe than illnesses caused by other gastrointestinal viruses [21]. Rotaviruses belong to the *Reoviridae* family, which possess a double-stranded RNA genome, codifying for six structural (VP) and six non-structural (NSP) proteins. NSP1 is associated with the degradation of IFN type I precursors, which downregulates the cellular antiviral response, improving viral replication that associates with diarrhea severity [20,22,23].

The cellular immune response against pathogens is mediated by interferons (IFNs), a group of secreted cytokines that elicit distinct antiviral effects [24,25]. The host innate immune system relies on pathogen sensors. One of them is the retinoic acid-inducible gene-I (RIG-I)-like [24]. In viral infections, RIG-I is essential for antiviral defense and type I interferon induction, through the interferon regulatory factors (IRFs) IRF-3, IRF-5, and IRF-7. Type I interferons are regulated by IFN-α and IFN-β genes and are directly induced in antiviral responses [24,26,27]. Nevertheless, some viruses express proteins associated with cellular immunity evasion by inhibiting the antiviral type I interferon pathway [7,22,23]. Some probiotics such as *Lactobacillus mucosae* and *Bifidobacterium breve* have been shown to restore antiviral signaling by upregulating interferon levels [7,28].

Although the properties of probiotics and prebiotics on human health are well recognized, recent research on microalgae has shown their potential to modulate human immune response and act as antimicrobial agents against enteric pathogens [25]. We have previously demonstrated that *B. longum* and *C. sorokiniana* possess antiviral effects against rotavirus. The antiviral activity of *B. longum* and/or *C. sorokiniana* has been associated with rotavirus reduced infectivity [8]. However, their effect on cellular response against rotavirus has not yet been elucidated. Therefore, this study aimed to determine whether *B. longum* and/or *C. sorokiniana* modulate cellular antiviral response mediated by type I interferons and precursors in rotavirus-infected HT-29 cells.

## 2. Materials and Methods 

### 2.1. Probiotic

*Bifidobacterium longum* strain (ATCC^®^ 15707) was grown on MPT medium (2.5 g yeast extract, 0.50 g sodium chloride, 0.5 g L-cysteine hydrochloride, 0.05 g ascorbic acid, 0.25 g potassium phosphate dibasic, 0.15 g potassium phosphate monobasic, 0.124 mg ferric ammonium citrate, 10 g casein digest peptone, and 5.0 g glucose), as previously reported [29]. Microbial growth kinetics were then evaluated by turbidimetry once the exponential phase was identified. In addition, we determined the colony-forming units (CFU) per milliliter by serial dilutions on agar plates and the bacterial inoculum was adjusted to 1 × 10^6^ CFU/mL, depending on the assay.

### 2.2. Chlorella sorokiniana

The microalga *Chlorella sorokiniana* was collected in the San Juan River in Cadereyta, Nuevo León, México (25°31′44″ N–100°2′8″ W). It was identified and subjected to phenotypic and genotypic characterization, as previously reported [9,30]. *C. sorokiniana* was grown in L-carnitine (LC) nutrient solution (5 mM KNO_3_, 1 mM KH_2_ PO_4_, 2 mM MgSO_4_·7H_2_O, 6.25 mM Ca(NO_3_)_2_·4H_2_O, 46 μM H_3_BO_3_, 9.15 μM MnCl_2_·4H_2_O, 765 nM ZnSO_4_·7H_2_O, 320 nM CuSO_4_·5H_2_O, 15 nM (NH_4_)6Mo_7_O_24_·4H_2_O, 20 μM FeSO_4_·7H_2_O, and 20 μM Na_2_ EDTA) at 25 °C and 120 rpm continuous light at 1400 lumens for 12 d [31].

### 2.3. Cells

We used rhesus monkey kidney cells (MA104; ATCC CRL-2378) in rotavirus replication assays, whereas HT-29 human colon tumor cells (ATCC HTB-38) were used in *B. longum* and *C. sorokiniana* assays. The cell line HT-29 was originally obtained in 1972 from a 44-year-old Caucasian female of blood group A, Rh-positive [32]. Cell lines were incubated in RPMI-1640 culture medium (Gibco, Grand Island, NY, USA), supplemented with 5% fetal bovine serum (FBS; Mediatech Inc., Corning, NY, USA), 2 mM L-glutamine, and 1% antibiotic and antimycotic solution (Caisson Laboratories, Smithfield, UT, USA) at 37 °C and 5% CO_2_, until confluence. Cells were then harvested with PBS and 0.25% trypsin (Mediatech Inc.) and transferred to 6-well plates for *B. longum*, *C. sorokiniana*, and rotavirus assays or 96-well microplates for rotavirus microtitration.

### 2.4. Rotavirus Strain and Viral Titration

The human group A rotavirus strain Wa was propagated in MA104 cells. Virus infection was activated by incubating at 37 °C in 5% CO_2_ for one hour with 10 µg/mL trypsin-EDTA solution 10× (Sigma-Aldrich, St. Louis, MO, USA), after which the inoculum was replaced with RPMI-1640 culture medium (Gibco) and incubated at 37 °C in 5% CO_2_ for 24 h. Lysates were then stored at −20 °C, until use. Rotavirus titers were calculated as focus-forming units per milliliter (FFU/mL) by immunochemistry, as previously reported [33]. In brief, lysates from virus propagation or cells infected with rotavirus and treated with probiotics and the microalga were used to infect MA104 in 96-well plates. After 14 h of incubation, cells were fixed with a cold acetone-PBS solution and incubated for 45 min at room temperature. Next, monolayer was washed twice with PBS and primary anti-rotavirus antibodies (Invitrogen, Carlsbad, CA, USA) were added. Cells were then incubated one hour at 37 °C and washed twice with PBS, after which horseradish peroxidase (HRP)-anti-sheep IgG conjugate (Invitrogen) was added to the cells and incubated one hour at 37 °C, followed by the addition of 0.1 M sodium acetate buffer (30 mM sodium acetate and 12 mM acetic acid; pH 5.0), containing 0.64 mg/mL aminoethylcarbazole (Sigma-Aldrich) dissolved in N,N-dimethyl formamide (Sigma-Aldrich) plus 0.36% hydrogen peroxide immediately prior to use. Infected cells were counted using optical microscopy and identified by their brown color, indicating the presence of viral antigens. FFU/mL were calculated using the following formula: FFU/mL = 20 × (microscope objective) × 5.5 (well diameter) × average number of foci (duplicate determinations; 100 to 200 foci/well) × dilution (foci count). Multiplicity of infection (MOI), which refers to the number of viral particles per cell, was calculated with the number of viral particles used (FFU/mL) per well divided by the number of cells originally seeded in the well. Rotavirus multiplicity of infection (MOI) was 0.1 in each assay.

### 2.5. Cellular Viability Assay

Viability of HT-29 cells treated with *B. longum* and/or *C. sorokiniana* was determined by the colorimetric MTT reduction assay, as previously reported [19]. For this, HT-29 cells were incubated with *C. sorokiniana* and/or *B. longum* in RPMI-1640 medium without FBS for 24 h at 37 °C and 5% CO_2_ in 95% air, after which they were washed twice with PBS and 20 µL of 3-(4, 5-dimethylthiazol-2-yl)-2, 5-diphenyltetrazolium bromide (MTT; Sigma-Aldrich; 5 mg/mL final concentration) were added to cells and incubated for 3 h. MTT was then replaced by 10 µL of dimethyl sulfoxide (DMSO; Sigma-Aldrich) and incubated three minutes, under continuous shaking. Next, optical densities were determined in a microplate reader (Multiskan GO, Thermo Fisher Scientific Inc., San Jose, CA, USA) at 570 nm [34].

### 2.6. Bifidobacterium longum and Rotavirus Assays

*B. longum* was used to treat rotavirus-infected cells. In pre-infection experiments, HT-29 cells were treated with *B. longum* (1 × 10^6^ CFU/mL) for one hour at 37 °C and 5% CO_2_ in 95% air, after which they were infected with Wa strain rotavirus (MOI 0.1) for 24 h at 37 °C and 5% CO_2_ in 95% air. In post-infection assays, HT-29 cells were infected with rotavirus (MOI 0.1) for one hour at 37 °C and 5% CO_2_ in 95% air and incubated with *B. longum* (1 × 10^6^ CFU/mL) in RPMI-1640 medium without FBS at 37 °C in 5% CO_2_ for 24 h. Cells were then stored at 20 °C, until further analysis, such as viral RNA purification and qPCR assays. *B. longum* cytotoxicity to HT-29 cells was determined by the MTT reduction assay.

### 2.7. Chlorella sorokiniana and Rotavirus Assays

*C. sorokiniana* biomass was used to treat rotavirus-infected cells. In pre-infection treatments, HT-29 cells were treated with *C. sorokiniana* (1 × 10^6^ cells/mL) for one hour at 37 °C and 5% CO_2_ in 95% air, after which they were infected with Wa strain rotavirus (MOI 0.1) for 24 h at 37 °C and 5% CO_2_ in 95% air. In post-infection assays, HT-29 cells were infected with rotavirus (MOI 0.1) for one hour at 37 °C and 5% CO_2_ in 95% air, and incubated with *C. sorokiniana* (1 × 10^6^ cells/mL) in RPMI-1640 medium without FBS at 37 °C in 5% CO_2_ for 24 h. Cells were then stored at 20 °C, until viral titration by immunochemistry or until further analysis, such as viral RNA purification and qPCR assays. Microalga cytotoxicity to HT-29 cells was determined by the MTT reduction assay.

### 2.8. Effect of Probiotic and Microalga Combination Treatment on Rotavirus-Infected Cells

*B. longum* and *C. sorokiniana* assays were performed on HT-29 cells, before and after rotavirus infection as follows: (a) in pre-infection experiments, before rotavirus infection, cells were incubated with *B. longum* (1 × 10^6^ UFC/mL) and *C. sorokiniana* (1 × 10^6^ cells/mL) in two milliliters of RMPI-1640 medium for 4 h at 37 °C and 5% CO_2_, after which cells were washed with PBS, followed by rotavirus infection (MOI 0.1) and incubation for 24 h at 37 °C and 5% CO_2_, and (b) in post-infection experiments, after rotavirus infection (1 h) at 4 h at 37 °C and 5% CO_2_, infected HT-29 cells were treated with *B. longum* (1 × 10^6^ UFC/mL) in combination with *C. sorokiniana* (1 × 10^6^ cells/mL) and incubated for 24 h at 37 °C and 5% CO_2_.

### 2.9. mRNA Purification and qPCR Assay

Total RNA extraction from rotavirus-infected cell lysates and/or treated with *C. sorokiniana* and/or *B. longum* was performed by the Trizol method (Life Technologies, Rockville, MD, USA). Purified RNA was used as a template for cDNA synthesis (High-Capacity cDNA Reverse Transcription; Applied Biosystems, Foster City, CA, USA). Relative expression of IFN-α, IFN-β, IRF-3, IRF-5, and RIG-I genes was determined by qPCR, using PGK-1 as an endogenous gene (Table 1). Reactions were developed with the Sensi FAST SYBER Lo-ROX Kit (Bioline, London, UK), following manufacturer’s instructions. qPCR conditions were 95 °C for 5 min, 45 cycles of 58 °C for 5 s, and 60 °C for 10 s. Gene relative expression was calculated using 2^−ΔΔCt^ (Applied Biosystems).

#### Statistical Analysis

Data were reported as mean ± SD of triplicates from three independent experiments. Statistical analysis was calculated by the one-way ANOVA and Tukey’s multiple comparisons or Kruskal–Wallis and Dunn’s multiple comparison tests, using GraphPad Prism version 9.5.1 (528), 24 January 2023 (GraphPad Software Inc., San Diego, CA, USA). *p* values < 0.05 were considered statistically significant.

## 3. Results

### 3.1. Pre-Infection or Post-Infection Assays with Rotavirus in HT-29 Cells

To study the effect of *B. longum* and *C. sorokiniana* on rotavirus infection, HT-29 cells were treated with one or both of them and infected with rotavirus or they were first infected and treated. As shown in Figure 1, all treatments (pre-infection or post-infection treatments with rotavirus) improved the monolayer integrity, as compared with rotavirus-infected cells without any treatment.

### 3.2. Rotavirus

To study the cellular antiviral response induced by rotavirus in infected cells without any treatment, we determined mRNA relative expression level of IFN-α, IFN-β, IRF-3, IRF-5, and RIG-I. Results showed an increased relative expression of IFN-β (*p* < 0.05) and IRF-5 (*p* < 0.05). In contrast, IFN-α, IRF-3, and RIG-1 genes relative expression was lower as compared with that of untreated or infected cells.

### 3.3. Bifidobacterium longum and Rotavirus

In pre- and post-infection experiments, results on rotavirus-infected cells treated with *B. longum* showed better monolayer integrity, compared with that of rotavirus-infected cells without the probiotic (Figure 1). In addition, to determine whether *B. longum* induces an in vitro antiviral response in HT-29 cells, the effect of this probiotic in pre- and post-infected cell experiments was measured through the mRNA relative expression levels of IFN-α, IFN-β, IRF-3, IRF-5, and RIG-I genes. In pre-infection assays, HT-29 cells were treated with *B. longum* before rotavirus infection, showing significant (*p* < 0.05) increases in IFN-α and RIG-I relative genes expression, as compared with infected cells without treatment. In post-infection experiments, cells were infected and further treated with probiotics. Results indicated downregulation of IRF-3 and IRF-5, without statistically significant difference with rotavirus alone. In contrast, in pre- and post-infection experiments, higher relative expression was observed in RIG-I, as compared with infected cells without *B. longum* (Figure 2). Furthermore, there was a significantly (*p* < 0.05) higher relative expression of IFN-α in pre-infection experiments than that in post-infection ones. However, not significant differences were observed among IFN-β, IRF-5, and RIG-1. This result demonstrated that *B. longum* may induce an in vitro antiviral response in HT-29 cells, mediated by IFN-α in cells treated with this probiotic and further infected with rotavirus (Figure 2).

### 3.4. Anti-Rotavirus Effect of Chlorella sorokiniana in HT-29 Cells

Results on rotavirus-infected cells treated with *C. sorokiniana*, in pre- and post-infection experiments, showed better monolayer integrity than that in cells infected and without the microalgae treatment (Figure 1). To determine if *C. sorokiniana* induces an in vitro antiviral response in HT-29 cells, we measured IFN-α, IFN-β, IRF-3, IRF-5, and RIG-I mRNA relative expression levels. In pre-infection experiments, cells were treated with *C. sorokiniana* followed by rotavirus infection, showing that IFN-α, IRF-3, and RIG-I relative expression was significantly (*p* < 0.05) upregulated, as compared with that in rotavirus-infected cells without treatment. In post-infection experiments, cells were infected and treated with *C. sorokiniana,* showing a lower relative expression of IFN-α, IFN-β, IRF-3, and RIG-1 than that in cells only infected with rotavirus. In addition, IFN-α, IRF-3, and RIG-I relative expression was significantly (*p* < 0.05) higher in pre-infection experiments than in post-infection ones. This may indicate that the antiviral response induced by *C. sorokiniana* is mediated by RIG-I, IRF-3, and IFN-α in pre-infection experiments (Figure 3).

### 3.5. Bifidobacterium longum, Chlorella sorokiniana, and Rotavirus

HT-29 cells treated with the *B. longum* and *C. sorokiniana* combination and infected with rotavirus, in pre- and post-infection experiments, showed an improved cell monolayer integrity, as compared with rotavirus-infected cells without treatment (Figure 1). Furthermore, to determine if *B. longum* and *C. sorokiniana* induced an in vitro antiviral response, we evaluated the mRNA relative expression level of IFN-α, IFN-β, IRF-3, IRF-5, and RIG-I genes. In pre-infection experiments, cells were treated with a combination of *B. longum* and *C. sorokiniana* before rotavirus infection, showing a higher relative expression of IFN-α and RIG-I than those in cells not treated but infected (*p* < 0.05). In post-infection experiments, cells were infected and further treated with *B. longum* in combination with *C. sorokiniana,* showing a significantly (*p* < 0.05) increased IFN-β relative expression, as compared with cells infected with rotavirus but without probiotic and microalgae treatments. In addition, IRF-3 and IRF-5 expression was significantly (*p* < 0.05) higher in cells treated pre-infection than in post-infection experiments. These results may indicate that the protective antiviral response induced by *B. longum* and *C. sorokiniana* is mediated by IFN-α in pre-infection assays and by IFN-β in post-infection treatments (Figure 4).

## 4. Discussion

Probiotics are, by definition, living microorganisms that confer health benefits to the host, if they are consumed in adequate amounts [38]. In this regard, probiotics, such as *Bifidobacterium* species, have been widely studied and used because they have effectively treated or prevented gastrointestinal infections, particularly viral infections [39,40,41]. Probiotics have been associated with a reduction of viral load and diarrhea duration in viral gastroenteritis [41,42]. In the present study, we performed in vitro experiments with rotavirus and probiotics in HT-29 cells, which exhibit similar structural and functional features than enterocytes [43]. The first insight into the favorable effect of *B. longum* against rotavirus was the monolayer integrity, compared with infected cells without any treatment. Other studies with *B. adolescentis* and *L. casei* showed a delayed cytopathic effect in MA104 cells infected with rotavirus [40]. In addition, *B. longum* R0175 was recently associated with the prevention of rotavirus infection and showed a protective effect on porcine intestinal epithelial cells before infection [42].

In infected cells, rotavirus induces the proteasomal degradation of IRF-3, IRF-5, and IRF-7 through NSP1 protein activity, suppressing the host’s antiviral interferon response [23,44]. In the present study, we observed that after 24 h of infection with rotavirus in HT-29 cells, the relative expression of IRF-5 and IFN-β was significantly (*p* < 0.05) increased, as compared with non-infected cells. Furthermore, we have previously reported that *B. longum* reduced rotavirus infectivity to 74% [9]. Due to the protective effect of *B. longum* of cell monolayers and the potential effect of blocking rotavirus infectivity, we hypothesized that this probiotic may induce an antiviral cellular immune response against rotavirus.

Therefore, we determined whether *B. longum* modulates an antiviral response in in vitro experiments, before or after rotavirus infection. We observed a significantly (*p* < 0.05) higher IFN-α relative expression in pre-infection experiments than in post-infection ones. Previous studies indicated that IFN-α genes are induced in response to viral infection [23]. Our results indicate that *B. longum* anti-rotavirus activity may be mediated by blocking virus infectivity (as we previously reported) and inducing a cellular antiviral response (present study). In agreement with our work, others have demonstrated that the innate immune response against rotavirus was modulated by *B. infantis* MCC12 and *B. breve* MCC1274 in porcine intestinal epitheliocyte cells [6]. An antiviral cellular response might be associated with molecules such as lipopolysaccharides or lipoteichoic acids produced by probiotics that may induce antiviral gene expression, associated with shorter infection periods and a lower risk of viral infection [39,45].

On the other hand, microalgae have demonstrated significant health-promoting effects and, in some cases, function as prebiotics. Cells treated with *C. sorokiniana* and infected with rotavirus (pre-infection experiments) showed increased RIG-1 and IFN-α relative expression, as compared with rotavirus-infected cells (without treatment). Furthermore, the combination of *B. longum* and the microalgae *C. sorokiniana* improved the monolayer integrity in rotavirus-infected cells, as compared with infected cells without any treatment. However, as we previously reported, the anti-rotavirus effect of *B. longum* was better in combination with *C. sorokiniana* because they reduced viral infectivity to 30% instead of 74% with the probiotic alone [9]. Due to the protective effect of *B. longum* and *C. sorokiniana* of cell monolayers, we evaluated whether they induce an antiviral cellular response against rotavirus. Our results indicated that a combination of *B. longum* and *C. sorokiniana* increased IFN-α levels pre-infection (*p* < 0.05) and IFN-β in experiments post-infection (*p* < 0.05). The anti-rotavirus effect of *B. longum* combined with *C. sorokiniana* may be associated with blocking infectivity and inducing a cellular antiviral response. In this regard, we showed three-fold increases in RIG-I and IFN-α relative expression levels in *Bifidobacterium longum* and *Chlorella sorokiniana*-treated HT-29 cells, which may indicate a protective effect by preventing pathogenic infection, as previously reported for *Bifidobacteria* spp. [40,42]. On the other hand, in rotavirus-infected cells, the mRNA relative expression of IFN-α was suppressed (*p* < 0.05), whereas that of IRF-5 and IFN-β (*p* < 0.05) resulted in two-fold significant (*p* < 0.05) increases, as compared with non-infected cells. Furthermore, we observed a 17-fold increase in IFN-α relative expression in cells treated with *Bifidobacterium longum* and *Chlorella sorokiniana* and further infected with rotavirus, whereas in cells infected and further treated with *Bifidobacterium longum* and *Chlorella sorokiniana*, we showed a five-fold increase in IFN-β relative expression, as compared with non-infected cells. Our results indicated a better antiviral response in pre-treatment experiments and the influence of *Bifidobacterium longum* and/or *Chlorella sorokiniana* to induce an IFN type I-mediated antiviral response in rotavirus-infected cells (Figure 3 and Figure 4).

In addition to the anti-rotavirus activity shown in our studies, *Chlorella* supplementation was associated with a protective effect against chronic hepatitis C virus, in a clinical trial model, and was related to a reduced viral load [46]. Some microalgal cell components and chemical agents may exert health-promoting effects [47]. Although numerous studies regarding the antiviral effect of probiotics and the interaction of probiotics have been reported, the specific immune mechanism remained unknown [48]. Probiotics’ antiviral mechanism is still under research; however, microalgae studies are scarce. In addition to the antiviral activities of probiotics, some foods may enhance the energy sources of probiotics and might be used as prebiotics to potentiate their antiviral effect. In fact, by feeding rice bran and probiotics to germ-free swine, the intestinal barrier function was increased by the produced metabolites, regulating the immune response and preventing rotavirus-induced diarrhea [30]. This information leads us to believe that *Chlorella* activity may be like a prebiotic.

The anti-rotavirus effect of *B. longum* mediated by type I interferons reported in this study agrees with previous reports of probiotics against viruses and it has been associated with viral load reduction and improved monolayer integrity. In addition, our results indicated that *B. longum*, *C. sorokiniana*, or their combination improve cellular antiviral immune response. Furthermore, we showed that a combination of *B. longum* and *C. sorokiniana* significantly increased interferon response in pre-infection and post-infection experiments, as compared with individual effects. To our knowledge, this is the first report of the immunomodulatory effect of *B. longum* and *C. sorokiniana* in rotavirus-infected cells.

## 5. Conclusions

Our present results indicate that *Bifidobacterium longum* in combination with *Chlorella sorokiniana* modulates type I interferons relative expression, improving antiviral response. Although more studies are needed, the use of probiotics such as *Bifidobacterium* and the microalgae *Chlorella* may become an alternative to prevent rotavirus infection or decrease gastroenteritis severity.

## Figures and Tables

**Figure 1 microorganisms-11-01237-f001:**
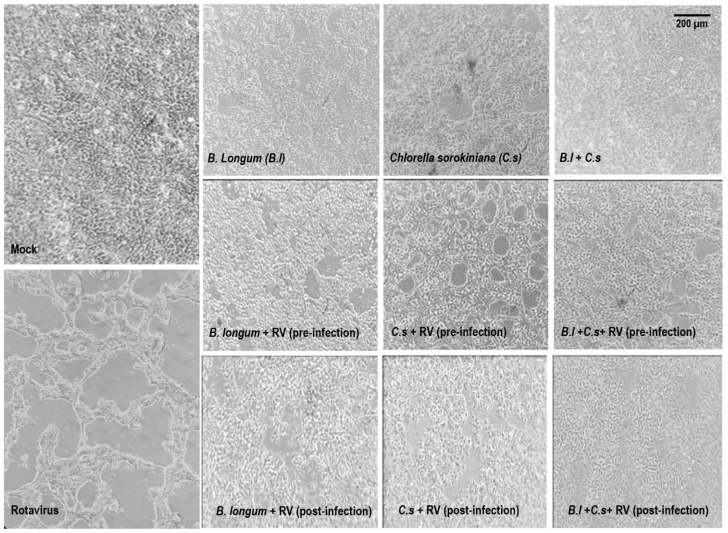
Monolayer integrity of rotavirus-infected cells at 24 h post-infection. In pre-infection experiments, HT-29 cells were incubated with *Bifidobacterium longum* and/or *Chlorella sorokiniana*, after which they were infected with rotavirus. In post-infection experiments, HT-29 cells were first infected with rotavirus and further treated with *Bifidobacterium longum* and/or *Chlorella sorokiniana*. Controls included mock (complete monolayer) and rotavirus (absence of monolayer). Abbreviations: *B.l*: *Bifidobacterium longum*; *C.s*: *Chlorella sorokiniana*; and RV: Rotavirus.

**Figure 2 microorganisms-11-01237-f002:**
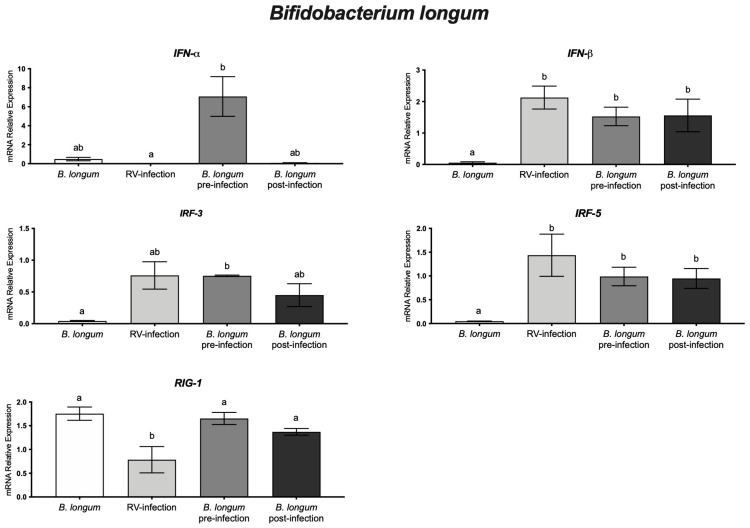
*Bifidobacterium longum* and rotavirus (RV) experiments. Relative expression level of IFN-α, IFN-β, IRF-3, IRF-5, and RIG-I genes in HT-29 cells treated with *B. longum* pre- and post- rotavirus infection. Different letters indicate statistical significance between treatments. Data were analyzed by ANOVA with subsequent Fisher test, using GraphPad Prism 9.5.

**Figure 3 microorganisms-11-01237-f003:**
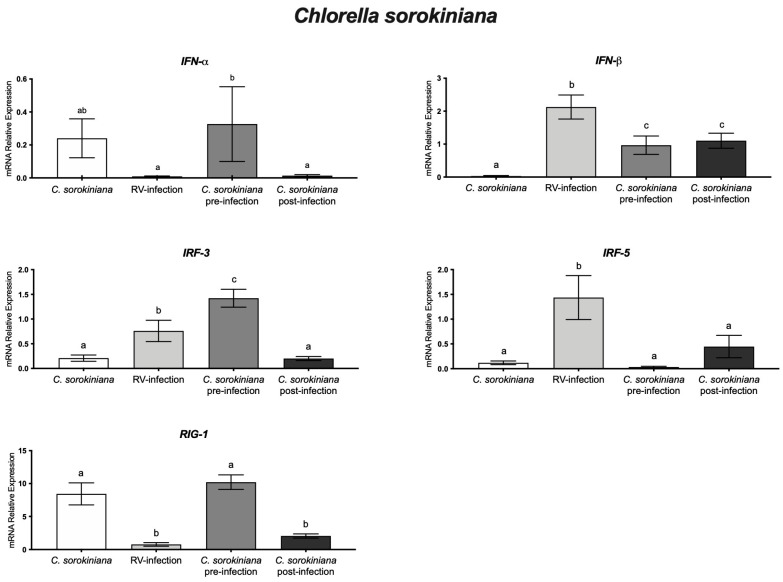
*Chlorella sorokiniana* and rotavirus (RV) experiments. mRNA relative expression level of IFN-α, IFN-β, IRF-3, IRF-5, and RIG-I genes in HT-29 cells treated with *Chlorella sorokiniana* pre- and post- rotavirus Wa infection. Different letters indicate statistical significance between treatments. Data were analyzed by ANOVA with subsequent Fisher test using GraphPad Prism 9.5.

**Figure 4 microorganisms-11-01237-f004:**
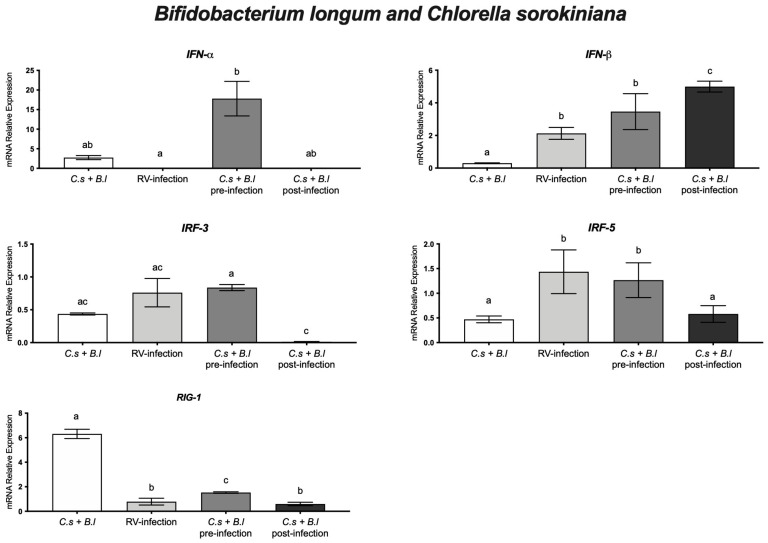
*Bifidobacterium longum* (*B.l*) and *Chlorella sorokiniana* (*C.s*) combination and rotavirus (RV) experiments. mRNA relative expression level of IFN-α, IFN-β, IRF-3, IRF-5, and RIG-I genes in rotavirus-infected HT-29 cells treated with *Bifidobacterium longum* in combination with *Chlorella sorokiniana*. Different letters indicate statistical significance between treatments. Data were analyzed by ANOVA with subsequent Fisher test using GraphPad Prism 9.5.

**Table 1 microorganisms-11-01237-t001:** Quantitative PCR primer sequences.

Primer Name	Primer Sequences (5′ to 3′)		Product Length	Reference
	Fwd	Rev		
IFN-α	5′-TTT CTC CTG CCT GAA GGA CAG-3′	5′-TCC ATG ATT TCT GCT CTG ACA-3′	373	[35]
IFN-β	5′-CTC CTC CAA ATT GCT CTC CTG-3′	5′-GCA AAC TGG TCA CGA ATT TTC C-3′	409	[35]
IRF-3	5′-ACC AGC CGT GGA CCA AGA G-3′	5′-TAC CAA GGC CCT GAG GCA C-3′	65	[36]
IRF-5	5′-CTG TCT CTG GTC TGG TCA GC-3′	5′-GCC AGC CAG GTG AGT GTT TA-3′	564	[35]
RIG-1	5′-CTC CCG GCA CAG AAG TGT-3′	5′-CCT CTG CCT CTG GTT TGG-3′	170	[35]
PGK1	5′-GAG ATG ATT ATT GGT GGT GGA A-3′	5′-AGT CAA CAG GCA AGG TAA TC-3′	160	[37]

## Data Availability

The data presented in this study are available on request from the corresponding author.

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
