# Peer review of "Bifidobacterium longum and Chlorella sorokiniana Improve the IFN Type I-Mediated Antiviral Response in Rotavirus-Infected Cells"

_microorganisms, 2023, doi:10.3390/microorganisms11051237_

Round 1

Reviewer 1 Report

The study "Type I interferons-mediated enhanced anti-rotavirus activity by 2 Bifidobacterium longum and Chlorella sorokiniana" demonstrates the immune enhancing effects of Bifidobacterium longum and Chlorella sorokiniana, after going through the manuscripts, I have the following concerns for the authors

1- Figure 1, the photos are not that clear at all, we need better resolution photos and please add a scale bar and include controls treated with Bifidobacterium longum and Chlorella sorokiniana to investigate effects on cells

2- You have to define the abbreviations for the first time you mention so define the abbreviation in Figure 1 in the figure legends 

3- qPCR data are not easily digested or not well described, for example levels of immune response markers are going down  in case of treatment with Bifidobacterium longum and/or Chlorella sorokiniana compared to RV infection ( then the increase in the expression is because of the infection not the probiotic), please verify that what is the role of the probiotic in the immune response after the RV infection if the markers are already expressed due to the infection)

4- I would recommend changing the title of the manuscript to fit the described results 

5- Language and grammar check required 

6- Define probiotic much better in the introduction, you can make use of this ref. doi: 10.3390/ph14040341

Language and grammar check is required 

Author Response

Dear REVIEWER 1

Thank you for your comments, the manuscript has been improved attending your suggestions. We have either replied to each specific comment addressed or incorporated your suggestions in the revised manuscript. We hope that this revision will answer the thoughtful and significant concerns.

Sincerely

The authors: Ricardo Romero-Arguelles, Patricia Tamez-Guerra, Guadalupe González-Ochoa, César I. Romo-Sáenz, Ricardo Gomez-Flores, Lilian Flores-Mendoza, Elizama Aros-Uzarraga.

IN RESPONSE TO THE COMMENTS:

The study "Type I interferons-mediated enhanced anti-rotavirus activity by 2 Bifidobacterium longum and Chlorella sorokiniana" demonstrates the immune enhancing effects of Bifidobacterium longum and Chlorella sorokiniana, after going through the manuscripts, I have the following concerns for the authors

1- Figure 1, the photos are not that clear at all, we need better resolution photos and please add a scale bar and include controls treated with Bifidobacterium longum and Chlorella sorokiniana to investigate effects on cells.

ANSWER:  We have prepared better resolution photos, added a scale bar, and included controls treated with Bifidobacterium longum and Chlorella sorokiniana, as suggested.  

2- You have to define the abbreviations for the first time you mention so define the abbreviation in Figure 1 in the figure legends.

ANSWER: We have defined the abbreviations for the first time they were mentioned, as suggested.

3- qPCR data are not easily digested or not well described, for example levels of immune response markers are going down  in case of treatment with Bifidobacterium longum and/or Chlorella sorokiniana compared to RV infection ( then the increase in the expression is because of the infection not the probiotic), please verify that what is the role of the probiotic in the immune response after the RV infection if the markers are already expressed due to the infection)

ANSWER: We have defined the role of the probiotic in the immune response after RV infection in the revised manuscript. Our results may indicate that B. longum anti-rotavirus activity may be mediated by blocking virus infectivity (as we previously reported) and inducing a cellular antiviral response (present study). Lines 255 to 262 and 349 to 354 of the revised manuscript.

4- I would recommend changing the title of the manuscript to fit the described results: Type I interferons-mediated enhanced anti-rotavirus activity by Bifidobacterium longum and Chlorella sorokiniana. Bifidobacterium longum and Chlorella sorokiniana improve the IFN type I-mediated antiviral response in rotavirus-infected cells. ???

ANSWER: We liked the title suggested and now it reads Bifidobacterium longum and Chlorella sorokiniana improve the IFN type I-mediated antiviral response in rotavirus-infected cells.

5- Language and grammar check required.

ANSWER: Language and grammar check was revised and edited by a professional.

6- Define probiotic much better in the introduction, you can make use of this ref. doi: 10.3390/ph14040341.

ANSWER: We have improved the Introduction section, as suggested. Lines 31 to 34 of the revised manuscript.

Reviewer 2 Report

The model of HT29 cells should be described in more detail, in the methods and in the intro and results sections. The use of special media for cultures should be justified and discussed, No other bacteria or microorganisms were evaluated for comparisons, so the study is observational, and no general conclusions can be made. Which rotavirus groups were used? There are no measures of cytopathic effects in control cultures infected, so the measures of mRNAs up or down regulated have no direct relationship to quantifiable antiviral effects (not specified throughout the text). The physiological response seems to be driven by IFN-beta, but most effects seen indicate changes of IFN-alpha RNA, no discussion is made. There are no measurements of the protein secretion of diferent IFN type I molecules, so changes in RNAs are just observations without effect on the final production of interferons. All teh study should be described in this light.

English use is correct and clear. Some typos are found (e.g. significate in lines 250 and 253, y in line 313, etc.)

Author Response

Dear REVIEWER 2

Thank you for your comments, the manuscript has been improved attending your suggestions. We have either replied to each specific comment addressed or incorporated your suggestions in the revised manuscript. We hope that this revision will answer the thoughtful and significant concerns.

Sincerely

The authors: Ricardo Romero-Arguelles, Patricia Tamez-Guerra, Guadalupe González-Ochoa, César I. Romo-Sáenz, Ricardo Gomez-Flores, Lilian Flores-Mendoza, Elizama Aros-Uzarraga.

IN RESPONSE TO THE COMMENTS:

The model of HT29 cells should be described in more detail, in the methods and in the intro and results sections.

ANSWER: We have described in more detail the HT29 model in the Methods and Introduction sections, as suggested. Lines 83, 106 to 108, and 324 to 325 of the revised manuscript.

The use of special media for cultures should be justified and discussed.

ANSWER:  The special media for cultures was cited in this work and previously described by our research team.

No other bacteria or microorganisms were evaluated for comparisons, so the study is observational, and no general conclusions can be made.

ANSWER: In previous assays, we evaluated several probiotics but focused on Bifidobacterium because it produced the best results in reducing rotavirus load (data not shown).

Which rotavirus groups were used?

ANSWER: We have included the specific strain of rotavirus in the revised manuscript (Human rotavirus group A, strain Wa), as suggested. Line 117 of the revised manuscript.

There are no measures of cytopathic effects in control cultures infected, so the measures of mRNAs up or down regulated have no direct relationship to quantifiable antiviral effects (not specified throughout the text).

ANSWER: We did not measure cytopathic effects. We only described the cells monolayer integrity of rotavirus-infected cells and with the probiotic. However, we have reported that Bifidobacterium longum reduced rotavirus infectivity to 74% and the combination of B. longum and C. sorokiniana reduced viral infectivity to 30% (Cantú-Bernal et al., 2020). In the present study, we evaluated the relative expression of type I interferons, considering the reduction of rotavirus infectivity associated with B. longum (viral load reduction determined by immunoperoxidase microtitration).

The physiological response seems to be driven by IFN-beta, but most effects seen indicate changes of IFN-alpha RNA, no discussion is made.

ANSWER: We have improved the Discussion section on this issue, as suggested. Line 352 of the revised manuscript.

There are no measurements of the protein secretion of different IFN type I molecules, so changes in RNAs are just observations without effect on the final production of interferons. All the study should be described in this light.

ANSWER: We attended your suggestion.

Comments on the Quality of English Language

English use is correct and clear. Some typos are found (e.g., significate in lines 250 and 253, y in line 313, etc.)

ANSWER: Typos were corrected, as suggested.

Reviewer 3 Report

Review of the article “Type I interferons-mediated enhanced anti-rotavirus activity by Bifidobacterium longum and Chlorella sorokiniana”

The search for approaches to the prevention and treatment of rotavirus infection is currently relevant. Rotavirus infection is still one of the most common infectious diseases of the gastrointestinal system in childhood. It is also the leading cause of morbidity and mortality from diarrhea in children.

The work is devoted to the study of the mechanisms of influence of Bifidobacterium longum in combination with Chlorella sorikinana on the antiviral response mediated by IFN-I type in cells infected with rotavirus. The authors used an original approach to the study by studying the gene expression of IFN alpha and beta and signaling molecules RIG-I, IRF-3 and IRF-5, which precede the formation and ensure the induction of interferons.

The presented material is scientifically substantiated, the research methods are described in detail and can be reproduced, the conclusions are consistent with the research data. The manuscript is well structured, the references are mostly adequate and refer to recent publications.

There are comments on the drawings.

Fig 1. In the caption to fig. it is necessary to clarify: this is lifelong observation or fixed preparations; and add the magnification level of the microscope. In addition, in the signature, indicate what is shown in the pictures - a complete monolayer, the absence of a monolayer, a sparse monolayer, the presence of viral degeneration loci. Separately give a transcript of the abbreviations.

Fig. 2. Errors in the figure by notation - mARN instead of mRNA; it is not clear what is denoted as "a"; "b"; "ab". Must be indicated in the caption to Fig. designations along the abscissa and ordinate axis.

Fig. 3. Remarks are similar for Fig.2.

Fig. 4. Remarks are similar for Fig.2. It is not clear what is denoted as "a"; "b"; "ab"; "With". It is necessary to indicate in the signature the abbreviation of the names Bifidobacterium longum and Chlorella sorokiniana.

Typos:

line 181: “PBS, followed by rotavirus infection (MOI 0.1) and incubation for 24 h for 4 h at 37 °C” ?

line 247 needs to be clarified “The results showed an increased genes expression levels of IFN” …

line 249: “The results indicated an gene expression IRF-3 and IRF-5 down-regulation without significate differences with rotavirus alone.”

Thus, the present study is original, consistent with the subject matter of the journal Microbiology. The conclusions are substantiated and supported by the results. The article is written properly and may be of interest to a wide range of readers.

The article must be accepted after minor changes.

Author Response

Dear REVIEWER 3

Thank you for your comments, the manuscript has been improved attending your suggestions. We have either replied to each specific comment addressed or incorporated your suggestions in the revised manuscript. We hope that this revision will answer the thoughtful and significant concerns.

Sincerely

The authors: Ricardo Romero-Arguelles, Patricia Tamez-Guerra, Guadalupe González-Ochoa, César I. Romo-Sáenz, Ricardo Gomez-Flores, Lilian Flores-Mendoza, Elizama Aros-Uzarraga.

IN RESPONSE TO THE COMMENTS:

The search for approaches to the prevention and treatment of rotavirus infection is currently relevant. Rotavirus infection is still one of the most common infectious diseases of the gastrointestinal system in childhood. It is also the leading cause of morbidity and mortality from diarrhea in children.

The work is devoted to the study of the mechanisms of influence of Bifidobacterium longum in combination with Chlorella sorikinana on the antiviral response mediated by IFN-I type in cells infected with rotavirus. The authors used an original approach to the study by studying the gene expression of IFN alpha and beta and signaling molecules RIG-I, IRF-3 and IRF-5, which precede the formation and ensure the induction of interferons.

The presented material is scientifically substantiated, the research methods are described in detail and can be reproduced, the conclusions are consistent with the research data. The manuscript is well structured, the references are mostly adequate and refer to recent publications.

There are comments on the drawings.

Fig 1. In the caption to fig. it is necessary to clarify: this is lifelong observation or fixed preparations; and add the magnification level of the microscope. In addition, in the signature, indicate what is shown in the pictures - a complete monolayer, the absence of a monolayer, a sparse monolayer, the presence of viral degeneration loci. Separately give a transcript of the abbreviations.

ANSWER: We have complied with this reviewer´s suggestions in the revised manuscript.

Fig. 2. Errors in the figure by notation - mARN instead of mRNA; it is not clear what is denoted as "a"; "b"; "ab". Must be indicated in the caption to Fig. designations along the abscissa and ordinate axis.

ANSWER: We have clarified this point by writing “different letters indicate statistical significance between treatments”. mARN now reads mRNA, as suggested.

Fig. 3. Remarks are similar for Fig.2.

ANSWER: We have clarified this point by writing “different letters indicate statistical significance between treatments”.

Fig. 4. Remarks are similar for Fig.2. It is not clear what is denoted as "a"; "b"; "ab"; "With". It is necessary to indicate in the signature the abbreviation of the names Bifidobacterium longum and Chlorella sorokiniana.

ANSWER: We have clarified this point by writing “different letters indicate statistical significance between treatments”. Figure 4 is the only that has these necessary abbreviation to match with the abbreviations written in the bars “Figure 4. Bifidobacterium longum (B.l) and Chlorella sorokiniana (C.s) combination and rotavirus (RV) experiments. mRNA relative expression level of IFN-a, IFN-b, IRF-3, IRF-5, and RIG-I genes in rotavirus-infected HT-29 cells treated with Bifidobacterium longum in combination with Chlorella sorokiniana. Different letters indicate statistical significance between treatments. Data were analyzed by ANOVA with subsequent Fisher test using GraphPad Prism 7.0. “.

Typos:

line 181: “PBS, followed by rotavirus infection (MOI 0.1) and incubation for 24 h for 4 h at 37 °C” ?

line 247 needs to be clarified “The results showed an increased genes expression levels of IFN” …

line 249: “The results indicated an gene expression IRF-3 and IRF-5 down-regulation without significate differences with rotavirus alone.”

ANSWER: Typos were corrected in the revised manuscript, as suggested.

Thus, the present study is original, consistent with the subject matter of the journal Microbiology. The conclusions are substantiated and supported by the results. The article is written properly and may be of interest to a wide range of readers.

The article must be accepted after minor changes.

ANSWER: We deeply appreciate this reviewer´s comment.

Round 2

Reviewer 1 Report

I would accept the manuscript for publication in its present form

Minor editing of English is required

Author Response

Dear Reviewer 1

Your comment was: Minor editing of English is required

ANSWER: We have complied with your suggestion.

Best regards

Reviewer 2 Report

I don't see all comments addressed, and no reasons given for not doing so. The points are the discrepancy between IFNs in natural rotavirus infection and those induced by pretreatment with probiotics; the meaning of additive effects of two organisms, without proper control; and the absence of quantification of the antiviral mediators.

with minor corrections, adequate

Author Response

Dear reviewer 2

Your comments were:

1. I don't see all comments addressed, and no reasons given for not doing so. The points are the discrepancy between IFNs in natural rotavirus infection and those induced by pretreatment with probiotics; the meaning of additive effects of two organisms, without proper control; and the absence of quantification of the antiviral mediators.

ANSWER: We comprehensively addressed the information in our manuscript (lines 372 to 385 of the revised manuscript)

2. Minor editing of English language required.

ANSWER: We have complied with your suggestion.

Best regards